# Carbon nanotubes as excitonic insulators

Daniele Varsano [1], Sandro Sorella[2], Davide Sangalli [3], Matteo Barborini[1,5], Stefano Corni [1,6], Elisa Molinari[1,4] & Massimo Rontani [1]

Fifty years ago Walter Kohn speculated that a zero-gap semiconductor might be unstable against the spontaneous generation of excitons–electron–hole pairs bound together by Coulomb attraction. The reconstructed ground state would then open a gap breaking the symmetry of the underlying lattice, a genuine consequence of electronic correlations. Here we show that this excitonic insulator is realized in zero-gap carbon nanotubes by performing first-principles calculations through many-body perturbation theory as well as quantum Monte Carlo. The excitonic order modulates the charge between the two carbon sublattices opening an experimentally observable gap, which scales as the inverse of the tube radius and weakly depends on the axial magnetic field. Our findings call into question the Luttinger liquid paradigm for nanotubes and provide tests to experimentally discriminate between excitonic and Mott insulators.

[1] CNR-NANO, Via Campi 213a, 41125 Modena, Italy. [2] SISSA & CNR-IOM Democritos, Via Bonomea 265, 34136 Trieste, Italy. [3] CNR-ISM, Division of Ultrafast Processes in Materials (FLASHit), Area della Ricerca di Roma 1, 00016 Monterotondo Scalo, Italy. [4] Dipartimento di Scienze Fisiche, Informatiche e Matematiche (FIM), Università degli Studi di Modena e Reggio Emilia, 41125 Modena, Italy. [5] Present address: Physics & Materials Science Research Unit, University of Luxembourg, 162a Avenue de la Faïencerie, 1511 Luxembourg, Luxembourg. [6] Present address: Dipartimento di Scienze Chimiche, Università degli Studi di Padova, Via Marzolo 1, 35131 Padova, Italy. Correspondence and requests for materials should be addressed to M.R. (email: massimo.rontani@nano.cnr.it)

Long ago Walter Kohn speculated that gray tin—a zero-gap semiconductor—could be unstable against the tendency of mutually attracting electrons and holes to form bound pairs, the excitons[1]. Being neutral bosoniclike particles, the excitons would spontaneously occupy the same macroscopic wave function, resulting in a reconstructed insulating ground state with a broken symmetry inherited from the exciton character[2–5]. This excitonic insulator (EI) would share intriguing similarities with the Bardeen–Cooper–Schrieffer (BCS) superconductor ground state[4,6–11], the excitons—akin to Cooper pairs—forming only below a critical temperature and collectively enforcing a quasi-particle gap. The EI was intensively sought after in systems as diverse as mixed-valence semiconductors and semimetals[12,13], transition metal chalcogenides[14,15], photoexcited semiconductors at quasi equilibrium[16,17], unconventional ferroelectrics[18], and, noticeably, semiconductor bilayers in the presence of a strong magnetic field that quenches the kinetic energy of electrons[19,20]. Other candidates include electron–hole bilayers[21,22], graphene[23–26], and related two-dimensional structures[27–33], where the underscreened Coulomb interactions might reach the critical coupling strength stabilizing the EI. Overall, the observation of the EI remains elusive.

Carbon nanotubes, which are rolled cylinders of graphene whose low-energy electrons are massless particles[34,35], exhibit strong excitonic effects, due to ineffective dielectric screening and enhanced interactions resulting from one dimensionality[36–39]. As single tubes can be suspended to suppress the effects of disorder and screening by the nearby substrate or gates[40–42], the field lines of Coulomb attraction between electron and hole mainly lie unscreened in the vacuum (Fig. 1a). Consequently, the interaction is truly long ranged and in principle—even for zero gap—able of binding electron–hole pairs close to the Dirac point in momentum space (Fig. 1b). If the binding energy is finite, then the ground state is unstable against the spontaneous generation of excitons having negative excitation energy, $\varepsilon_u < 0$. This is the analog of the Cooper instability that heralds the transition to the superconducting state—the excitons replacing the Cooper pairs.

Here we focus on the armchair family of zero-gap carbon nanotubes, because symmetry prevents their gap from opening as an effect of curvature or bending[43]. In this paper we show that armchair tubes are predicted to be EIs by first-principles calculations. The problem is challenging, because the key quantities controlling this phenomenon—energy band differences and exciton binding energies—involve many-body corrections beyond density functional theory (DFT) that are of the order of a few meV, which is close to the limits of currently available methods. In turn, such weak exciton binding reflects in the extreme spatial extension of the exciton wave function, hence its localization in reciprocal space requires very high sampling accuracy. To address these problems, we perform state-of-the-art many-body perturbation theory calculations within the GW and Bethe–Salpeter schemes[44]. We find that bound excitons exist in the (3,3) tube with finite negative excitation energies. We then perform unbiased quantum Monte Carlo simulations[45] to prove that the reconstructed ground state is the EI, its signature being the broken symmetry between inequivalent carbon sublattices—reminiscent of the exciton polarization. Finally, to investigate the trend with the size of the system, which is not yet in reach of first-principles calculations, we introduce an effective-mass model, which shows that both EI gap and critical temperature fall in the meV range and scale with the inverse of the tube radius. Our findings are in contrast with the widespread belief that electrons in undoped armchair tubes form a Mott insulator—a strongly correlated Luttinger liquid[46–52]. We discuss the physical origin of this conclusion and propose independent experimental tests to discriminate between excitonic and Mott insulator.

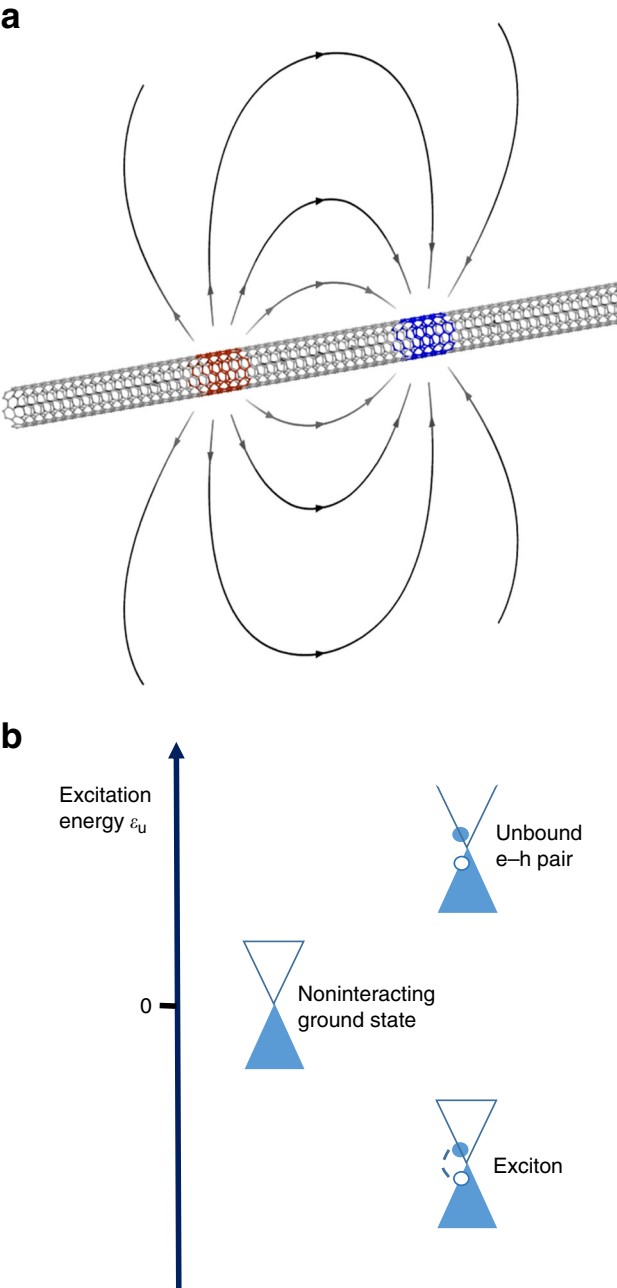

**Fig. 1** Excitonic instability in carbon nanotubes. **a** Sketch of a suspended armchair carbon nanotube. The field lines of the Coulomb force between electron and hole lie mainly in the vacuum, hence screening is heavily suppressed. **b** Excitonic instability in the armchair carbon nanotube. The scheme represents the excitation energy $\varepsilon_u$ of an electron–hole (e–h) pair relative to the noninteracting ground state, a zero-gap semiconductor. In the absence of interaction, the excitation energy $\varepsilon_u$ of an e–h pair is positive. The long-range interaction may bind e–h pairs close to the Dirac point in momentum space. If an exciton forms, then its excitation energy $\varepsilon_u$ is negative. This instability leads to the reconstruction of the ground state into an excitonic insulator

## Results

**Exciton binding and instability.** For the sake of computational convenience we focus on the smallest (3,3) armchair tube, which was investigated several times from first principles[53–60]. We first

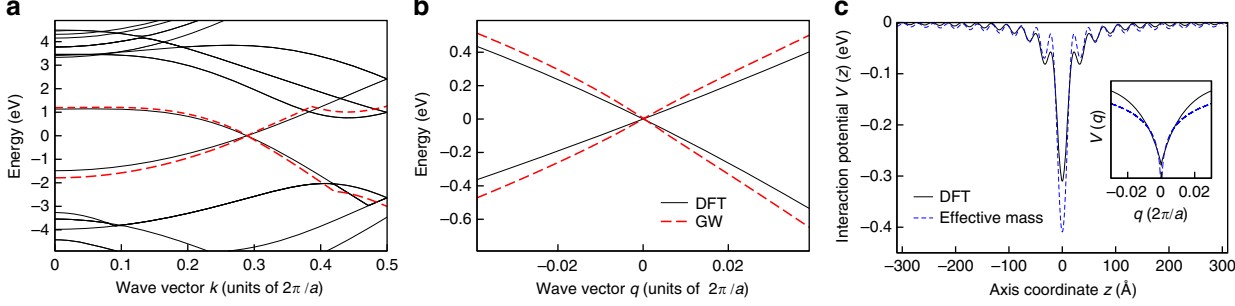

**Fig. 2** Electronic properties from many-body perturbation theory. **a** GW (dashed lines) and DFT (solid lines) band structure of the armchair carbon nanotube (3, 3). **b** Zoom close to the Dirac point K. The momentum $q$ is referenced from K. **c** Long-range part of electron–hole interaction $V(z)$ along the tube axis according to: DFT (solid line), effective-mass model (dashed line). Inset: interaction $V(q)$ in momentum space. $V$ is integrated over the mesh of the $q$ grid and projected onto the conduction and valence bands shown in panel b, with $|q| < 0.09(2\pi)/a$. The graphene lattice constant is $a=2.46$ Å

| Table 1 Excitation energies $\varepsilon_u$ of low-lying excitons of the (3, 3) tube obtained from first-principles many-body perturbation theory in units of meV | | |
| --- | --- | --- |
| | **Triplet** | **Singlet** |
| Lowest | −7.91 | −6.10 |
| 1st excited | −6.40 | −5.10 |
| 2st excited | 6.65 | 8.82 |

check whether the structural optimization of the tube might lead to deviations from the ideal cylindrical shape, affecting the electronic states. Full geometry relaxation (Methods) yields an equilibrium structure with negligible corrugation. Thus, contrary to a previous claim[60], corrugation cannot be responsible of gap opening. We find that the average length of C–C bonds along the tube axis, 1.431 Å, is shorter than around the circumference, 1.438 Å, in agreement with the literature[53].

We use DFT to compute the band structure (solid lines in Fig. 2a), which provides the expected[43] zero gap at the Dirac point K. In addition, we adopt the $G0W0$ approximation for the self-energy operator[44] to evaluate many-body corrections to Kohn–Sham eigenvalues. The highest valence and lowest conduction bands are shown as dashed lines. The zoom near K (Fig. 2b) shows that electrons remain massless, with their bands stretched by ∼28% with respect to DFT (farther from K the stretching is ∼13%, as found previously[56]). Since electrons and holes in these bands have linear dispersion, they cannot form a conventional Wannier exciton, whose binding energy is proportional to the effective mass. However, the screened e–h Coulomb interaction $V(z)$ along the tube axis, projected onto the same bands, has long range (Fig. 2c)—a remarkable effect of the topology of the tube holding even for vanishing gap. Consequently, $V(q)$ exhibits a singularity in reciprocal space at $q=0$ (smoothed by numerical discretization in the inset of Fig. 2c), which eventually binds the exciton. We solve the Bethe–Salpeter equation (BSE) over an ultradense grid of 1800 $k$-points, which is computationally very demanding but essential for convergence. We find several excitons with negative excitation energies $\varepsilon_u$, in the range of 1–10 meV (Table 1).

The exciton spectral weight is concentrated in a tiny neighborhood of K and K′ points in reciprocal space (Fig. 3b), hence the excitons are extremely shallow, spread over microns along the axis (Fig. 3c). Only e–h pairs with negative $k$ in valley K and positive $k$ in valley K′ contribute to the exciton wave function, which is overall symmetric under time reversal but not

under axis reflection within one valley, $k \to -k$, as shown in Fig. 3b (the axis origin is at Dirac point). On the contrary, the wave functions of excitons reported so far in nanotubes[36,37,56] are symmetric in $k$-space. The reason of this unusual behavior originates from the vanishing energy gap, since then e–h pairs cannot be backscattered by Coulomb interaction due to the orthogonality of initial and final states[61]. In addition, pair energies are not degenerate for $k \to -k$, as Dirac cones are slightly asymmetric (Supplementary Discussion and Supplementary Fig. 10).

The exciton with the lowest negative $\varepsilon_u$ makes the system unstable against the EI. The transition density, $\varrho_{tr}(\mathbf{r}) = \langle u|\hat{\varrho}(\mathbf{r})|0\rangle$, hints at the broken symmetry of the reconstructed ground state, as it connects the noninteracting ground state, $|0\rangle$, to the exciton state, $|u\rangle$, through the charge fluctuation operator $\hat{\varrho}$ (Fig. 3d). Here we focus on the simpler charge order (spin singlet excitons) and neglect magnetic phenomena (spin triplet), as the only relevant effect of spin–orbit coupling in real tubes[62,63] is to effectively mix both symmetries. Figure 3d may be regarded as a snapshot of the polarization charge oscillation induced by the exciton, breaking the inversion symmetry between carbon sublattices A and B. Note that this originates from the opposite symmetries of $|0\rangle$ and $|u\rangle$ under A ↔ B inversion and not from the vanishing gap. This charge displacement between sublattices is the generic signature of the EI, as its ground state may be regarded as a BCS-like condensate of excitons $|u\rangle$ (see the formal demonstration in Supplementary Note 5).

**Broken symmetry of the EI**. We use quantum Monte Carlo to verify the excitonic nature of the many-body ground state, by defining an order parameter characteristic of the EI, $\varrho_{AB}$. In addition, we introduce an alternative order parameter, $\varrho_{Transl}$, peculiar to a dimerized charge density wave (CDW) similar to the Peierls CDW predicted by some authors[57–59] for the smallest armchair tubes. The EI order parameter measures the uniform charge displacement between A and B sublattices, $\varrho_{AB} = \left(\sum_{i\in A} n_i - \sum_{i\in B} n_i\right)/N_{atom}$, whereas $\varrho_{Transl}$ detects any deviation from the periodicity of the undistorted structure by evaluating the charge displacement between adjacent cells, $\varrho_{Transl} = \sum_i n_i (-1)^{i_z}/N_{atom}$ (Fig. 4b–e). Here the undistorted structure is made of a unit cell of 12 C atoms repeated along the $z$ direction with a period of 2.445 Å and labeled by the integer $i_z$, $n_i$ is the operator counting the electrons within a sphere of radius 1.3 a.u. around the $i$th atom, and $N_{atom}$ is the total number of atoms in the cluster. Both order parameters $\varrho_{AB}$ and $\varrho_{Transl}$ vanish in the symmetric ground state of the undistorted structure, which

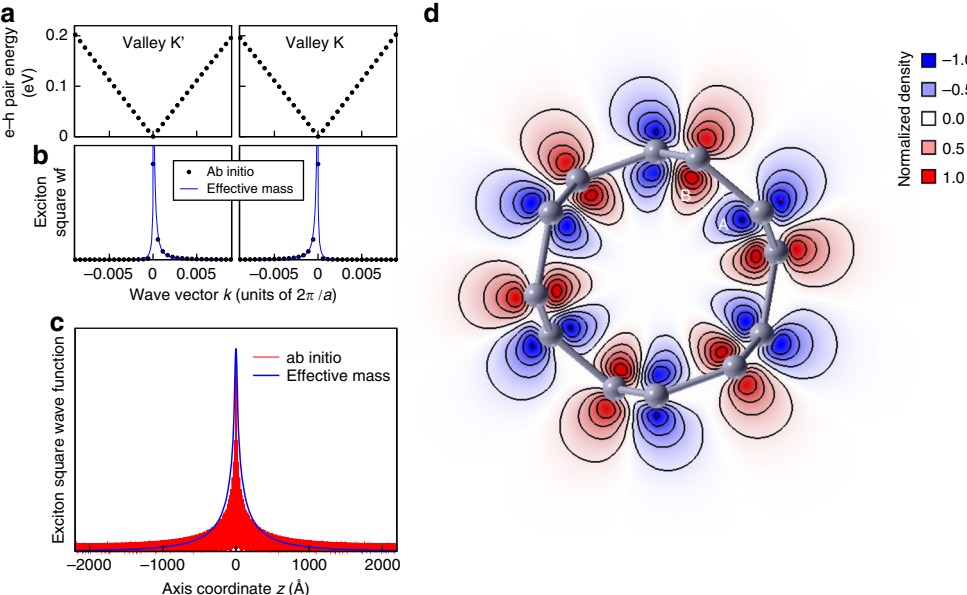

**Fig. 3** Wave function of the lowest-energy exciton of the (3, 3) tube. **a** GW excitation spectrum of free e–h pairs with zero center-of-mass momentum in the two Dirac valleys. **b** Square modulus of the triplet exciton wave function vs momentum $k$. Both first-principles (dots) and effective-mass (solid lines) probability weights accumulate asymmetrically close to Dirac points. The effective-mass model includes the dressed long-range interaction, the short-range intervalley exchange, and the small asymmetry of Dirac cones (cf. Supplementary Notes 1–3; a previous phenomenological theory[80] by one of the authors, which ignored the key role of long-range interaction, is ruled out by the present work). **c** Square modulus of the triplet exciton wave function vs e–h distance along the axis, $z$, according to first-principles (red curve) and effective-mass (blue curve) calculations. The Bohr diameter is larger than 2 μm. **d** Cross-sectional contour map of the transition density of the singlet exciton, $\varrho_{tr}(\mathbf{r})$, obtained from first principles. The blue/red color points to the deficit/surplus of charge, the isolines are equally spaced, the normalization of $\varrho_{tr}(\mathbf{r})$ is such that its maximum value is one, and letters label sublattices

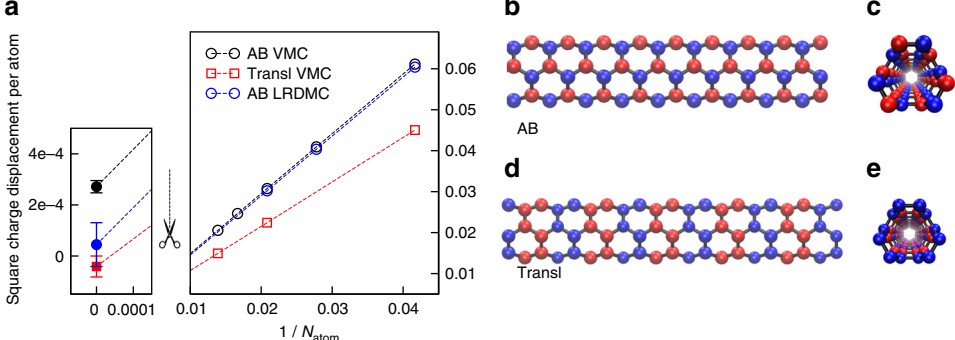

**Fig. 4** Broken symmetry of the ground state from quantum Monte Carlo. **a** The square of the charge displacement per atom (empty circles and squares for "AB" and "Transl" order parameters, respectively) is plotted vs the inverse of the number of atoms, $N_{atom}$, as obtained by variational (VMC) and lattice-regularized diffusion (LRDMC) quantum Monte Carlo. The filled symbols are linear extrapolations to the $N_{atom}=\infty$ limit. The error bars are estimated by means of the jackknife method using more than 30 independent samples for each independent twist (Methods). The error bars of empty symbols are not visible on the scale of the plot. **b–e** The sketches of the tube illustrate the two possible broken symmetries, with the blue/red color pointing to the deficit/surplus of charge. The AB order parameter, peculiar to the EI, is a uniform charge displacement between the two carbon sublattices (panels **b** and **c** show respectively the lateral and cross-sectional views of the tube). The Transl parameter is a charge displacement between two adjacent unit cells, signaling a charge density wave order breaking the translational symmetry (panels **d**, **e**)

is invariant under sublattice-swapping inversion and translation symmetries.

We then perform variational Monte Carlo (VMC), using a correlated Jastrow–Slater ansatz that has proved[64] to work well in 1D correlated systems (Methods), as well as it is able to recover the excitonic correlations present in the mean-field EI wave function[2–5] (Supplementary Discussion). We plot VMC order parameters in Fig. 4a. Spontaneously broken symmetry occurs in the thermodynamic limit if the square order parameter, either $\varrho_{AB}^2$ or $\varrho_{Transl}^2$, scales as $1/N_{atom}$ and has a non vanishing limit value for $N_{atom} \to \infty$. This occurs for $\varrho_{AB}^2$ (black circles in Fig. 4a),

confirming the prediction of the EI, whereas $\varrho_{Transl}^2$ vanishes (red squares), ruling out the CDW instability (see Supplementary Discussion as well as the theoretical literature[52,57–59] for the Peierls CDW case). We attribute the simultaneous breaking of sublattice symmetry and protection of pristine translation symmetry to the effect of long-range interaction.

The vanishing of $\varrho_{Transl}$ validates the ability of our finite-size scaling analysis to discriminate between kinds of order in the bulk. Though the value of $\varrho_{AB}$ after extrapolation is small, $\varrho_{AB} = 0.0165 \pm 0.0007$, it is non zero within more than twenty standard deviations. Besides, the quality of the fit of Fig. 4a

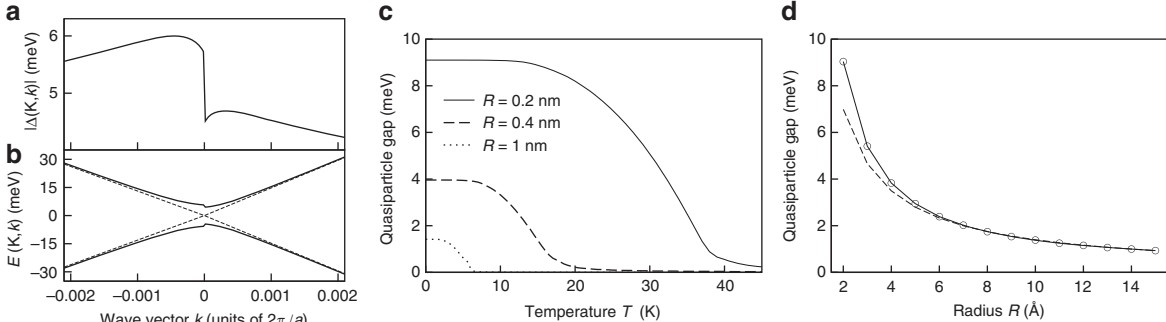

**Fig. 5** Excitonic insulator behavior from mean-field theory. **a** Excitonic order parameter, $|\Delta(\tau = \mathrm{K}, k)|$, vs momentum $k$ within K valley and **b** corresponding quasiparticle dispersion, $E(\mathrm{K}, k)$, for the (3, 3) armchair carbon nanotube. The data are derived by solving self-consistently the gap equation. For comparison, the noninteracting bands are indicated (dashed lines). The band in the K' valley is obtained by time reversal, as $|\Delta(\mathrm{K}', k)| = |\Delta(\mathrm{K}, -k)|$. **c** Quasiparticle gap vs temperature $T$ for different radii [for the (3, 3) tube $R=2$ Å]. **d** Quasiparticle gap vs $R$. The dashed curve is a fit proportional to $1/R$ pointing to the scaling behavior at large $R$

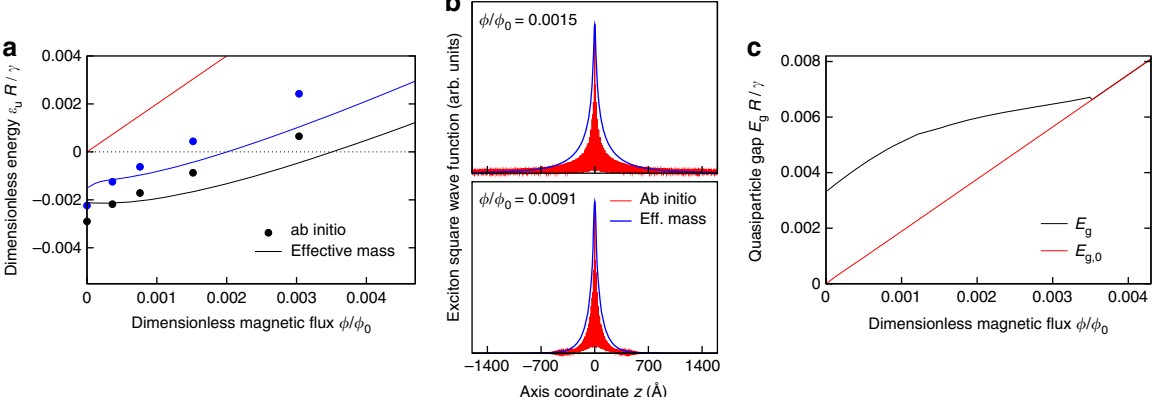

**Fig. 6** Effect of an axial magnetic field. **a** Excitation energies, $\varepsilon_u R/\gamma$, of low-lying excitons vs magnetic flux, $\phi/\phi_0$. Both first-principles (dots) and effective-mass (solid lines) data are reported. The black (blue) color labels the triplet (singlet) spin symmetry. The red line is the noninteracting gap and the dashed line is the instability threshold. **b** Square modulus of the wave function of the lowest exciton vs e–h distance along the axis, $z$, for increasing values of magnetic flux. Both ab initio (red lines) and effective-mass (blue lines) data are reported. **c** Total quasiparticle gap $E_g R/\gamma$ vs $\phi/\phi_0$. This observable may be accessed through Coulomb blockade spectroscopy. The red line is the noninteracting gap, $E_{g,0}$

appears good, because the data for the five largest clusters are compatible with the linear extrapolations of both $\varrho_{AB}^2$ and $\varrho_{Transl}^2$ within an acceptable statistical error. The more accurate diffusion Monte Carlo (LRDMC) values (obtained with the lattice regularization), shown in Fig. 4a as blue circles, confirm the accuracy of the variational calculation. However, as their cost is on the verge of present supercomputing capabilities, we were unable to treat clusters larger that $N_{atom}=48$, hence the statistical errors are too large to support a meaningful non-zero value in the thermodynamic limit. Nevertheless, we obtain a non zero LRDMC value smaller than the one estimated by VMC but compatible with it within a few standard deviations.

**Trends**. As the extension of our analysis to systems larger than the (3,3) tube is beyond reach, we design an effective-mass theory to draw conclusions about trends in the armchair tube family, in agreement with first-principles findings. We solve the minimal BSE for the massless energy bands $\varepsilon(k)=\pm\gamma|k|$ (Fig. 2b and Supplementary Note 1) and the long-range Coulomb interaction $V(q)$, the latter diverging logarithmically in one dimension for small momentum transfer $q$, $V(q)=(2e^2/A\kappa_r)\ln(|q|R)$ (inset of Fig. 2c and Supplementary Note 2). Here $\gamma$ is graphene tight-binding parameter including GW self-energy corrections, $k$ is the wave vector along the axis, $A$ is the tube length, $R$ is the radius,

and $\kappa_r$ accounts for screening beyond the effective-mass approximation. By fitting the parameters $\gamma=0.5449$ eV nm and $\kappa_r=10$ to our first-principles data, we obtain a numerical solution of BSE recovering ~60% of the lowest exciton energy $\varepsilon_u$ reported in Table 1 (Supplementary Note 3). Moreover, the wave function agrees with the one obtained from first principles (Fig. 3b, c). Importantly, $\varepsilon_u$ smoothly converges in an energy range that—for screened interaction—is significantly smaller than the extension of the Dirac cone, with no need of ultraviolet cutoff (Supplementary Fig. 9). Therefore, the exciton has an intrinsic length (binding energy), which scales like $R$ ($1/R$).

We adopt a mean-field theory of the EI as we expect the long-range character of excitonic correlations to mitigate the effects of quantum fluctuations. The EI wave function can be described as

$$|\Psi_{EI}\rangle = \prod_{\sigma\sigma'\tau k}\left[u_{\tau k} + \chi_{\sigma\sigma'}v_{\tau k}e^{i\eta}\hat{c}_{k,\sigma}^{\tau+}\hat{v}_{k,\sigma'}^{\tau}\right]|0\rangle. \quad (1)$$

Here $|0\rangle$ is the zero-gap ground state with all valence states filled and conduction states empty, the operator $\hat{c}_{k,\sigma}^{\tau+}$ $\left(\hat{v}_{k,\sigma}^{\tau+}\right)$ creates an electron in the conduction (valence) band with wave vector $k$, spin $\sigma$, valley $\tau = \mathrm{K}$ or K', $\eta$ is an arbitrary phase, and the $2 \times 2$ matrix $\chi_{\sigma\sigma'}$ discriminates between singlet and triplet spin symmetries of the e–h pair $\hat{c}_{k,\sigma}^{\tau+}\hat{v}_{k,\sigma'}^{\tau}|0\rangle$ (Fig. 1b). The positive

variational quantities $u_{\tau k}$ and $\nu_{\tau k}$ are the population amplitudes of valence and conduction levels, respectively, with $u_{\tau k}^2 + v_{\tau k}^2 = 1$. Whereas in the zero-gap state $u_{\tau k} = 1$ and $v_{\tau k} = 0$, in the EI state both $u_{\tau k}$ and $v_{\tau k}$ are finite and ruled by the EI order parameter $\Delta(\tau k)$, according to $u_{\tau k} v_{\tau k} = |\Delta(\tau k)|/2E(\tau k)$, with $E(\tau k) = \left[\varepsilon(\tau k)^2 + |\Delta(\tau k)|^2\right]^{1/2}$. The parameter $\Delta(\tau k)$ obeys the self-consistent equation

$$|\Delta(\tau k)| = \sum_{\tau' q} V^{\tau\tau'}(k, k+q) u_{\tau' k+q} v_{\tau' k+q}, \qquad (2)$$

which is solved numerically by recursive iteration (here $V$ includes both long- and short-range interactions as well as form factors, see Supplementary Note 4). As shown in Fig. 5a, in each valley $|\Delta(\tau k)|$ is asymmetric around the Dirac point, a consequence of the peculiar character of the exciton wave function of Fig. 3b. The electrons or holes added to the neutral ground state are gapped quasiparticle excitations of the EI, whose energy bands $\pm E(\tau k)$ are shown in Fig. 5b. The order parameter at the Dirac point, $|\Delta(\tau, k = 0)|$, is half the many-body gap. This gap is reminescent of the exciton binding energy, since in the ground state all electrons and holes are bound, so one needs to ionize an exciton-like collective state to create a free electron–hole pair. The gap strongly depends on temperature, with a low-temperature plateau, a steep descent approaching the critical temperature, and a milder tail (Fig. 5c). The gap approximately scales as $1/R$ for different tubes (circles in Fig. 5d): whereas at large $R$ such scaling is exact (cf. dashed curve), at small $R$ the gap is enhanced by short-range intervalley interaction (the decay of $\Delta$ will be mitigated if $\kappa_r$ is sensitive to $R$).

In experiments, many-body gaps are observed in undoped, ultraclean suspended tubes[65], whereas Luttinger liquid signatures emerge in doped tubes[35,43]. Though it is difficult to compare with the measured many-body gaps[65], as the chiralities of the tubes are unknown and the radii estimated indirectly, the measured range of 10–100 meV is at least one order of magnitude larger than our predictions. By doping the tube, we expect that the enhanced screening suppresses the EI order, quickly turning the system into a Luttinger liquid. We are confident that advances in electron spectroscopies will allow to test our theory.

The broken symmetry associated with the EI ground state depends on the exciton spin[5]. For spin singlet ($\chi_{\sigma\sigma'} = \delta_{\sigma\sigma'}$) and order parameter real ($\eta = 0, \pi$), $|\Psi_{EI}\rangle$ breaks the charge symmetry between A and B carbon sublattices. The charge displacement per electron, $\Delta e/e$, at each sublattice site is

$$\frac{\Delta e}{e} = \pm \cos\eta \frac{a}{A} \sum_{\tau k} \frac{|\Delta(\tau k)|}{2E(\tau k)}, \qquad (3)$$

where the positive (negative) sign refers to the A (B) sublattice (Supplementary Note 6). For the (3,3) tube this amounts to $\varrho_{AB} = 0.0068$, which compares well with Monte Carlo estimates of 0.0067 and 0.0165 from LRDMC and VMC, respectively. Note that assessing the energy difference between EI and zero-gap ground states is beyond the current capability of quantum Monte Carlo: the mean-field estimate of the difference is below $10^{-6}$ Hartree per atom, which is less than the noise threshold of the method ($10^{-5}$ Hartree per atom).

**Effect of magnetic field**. The EI is sensitive to the opening of a noninteracting gap, $E_{g,0}$, tuned by the magnetic field parallel to the tube axis, $B$. The ratio of the flux piercing the cross section, $\phi = \pi R^2 B$, to the flux quantum, $\phi_0 = ch/e$, amounts to an Aharonov–Bohm phase displacing the position of the Dirac point along the transverse direction[66], $k_\perp = (\phi/\phi_0)R^{-1}$. Consequently, $E_{g,0} = 2\gamma|k_\perp|$ is linear with $\phi/\phi_0$ (red line in Fig. 6a, c). Figure 6a

shows the evolution of low-lying singlet (blue lines) and triplet (black lines) excitons of the (3, 3) tube. In addition, we have implemented a full first-principles description of $B$ building on a previous method[67]. First-principles (circles) and model (solid lines) calculations show a fair agreement, which validates the effective-mass theory since all free parameters have been fixed at zero field. Here we rescale energies by $R/\gamma$ since we expect the plot to be universal, except for small corrections due to short-range interactions. Excitation energies obtained within the effective-mass model crossover from a low-field region, where $\varepsilon_u$ is almost constant, to a high-field region, where $\varepsilon_u$ increases linearly with $\phi/\phi_0$. Exciton wave functions are effectively squeezed by the field in real space (Fig. 6b), whereas in reciprocal space they loose their asymmetric character: the amplitudes become evenly distributed around the Dirac points (Supplementary Discussion and Fig. 11) and similar to those reported in literature[36,37,56]. At a critical flux $\phi_c/\phi_0 \approx 0.035$ the excitation energy $\varepsilon_u$ becomes positive, hence the tube exits the EI phase and $\Delta$ vanishes in a BCS-like fashion. We point out that the critical field intensity, $B_c \approx 460 \, T \cdot (R \, [\text{Å}])^{-2}$, is out of reach for the (3, 3) tube but feasible for larger tubes. The total transport gap, $E_g = \left(E_{g,0}^2 + 4|\Delta|^2\right)^{1/2}$, first scales with $\phi/\phi_0$ as $E_{g,0}$, then its slope decreases up to the critical threshold $\phi_c/\phi_0$, where the linear dependence on $\phi/\phi_0$ is restored (Fig. 6c). This behavior is qualitatively similar to that observed by Coulomb blockade spectroscopy in narrow-gap tubes close to the "Dirac" value of $B$, which counteracts the effect of $E_{g,0}$ on the transport gap, fully suppressing the noninteracting contribution[65].

## Discussion

The observed[65] many-body gap of armchair tubes was attributed to the Mott insulating state. The system was modeled as a strongly interacting Luttinger liquid with a gap enforced by short-range interactions[46,49], whereas the long tail of the interaction was cut off at an extrinsic, setup-dependent length[47,48,50–52]. This model thus neglects the crucial effect of long-range interaction, which was highlighted in Fig. 1: were any cutoff length smaller than the intrinsic exciton length, which is micrometric and scales with $R$, excitons could not bind.

Whereas armchair carbon nanotubes are regarded as quintessential realizations of the Luttinger liquid, since their low-energy properties are mapped into those of two-leg ladders[46], we emphasize that this mapping is exact for short-range interactions only. Among e–h pair collective modes with total momentum $q = 0$, Luttinger liquid theory routinely describes plasmons[68] but not excitons. Contrary to conventional wisdom, armchair tubes are EIs.

The excitonic and Mott insulators are qualitatively different. The EI exhibits long-range charge order, which does not affect the translational symmetry of the zero-gap tube. In the Mott insulator, charge and spin correlations may or may not decay, but always add a $2\pi/(2k_F)$ [or $2\pi/(4k_F)$] periodicity to the pristine system, $k_F$ being the Fermi wave vector[50,51]. The EI gap scales like $1/R$ (Fig. 5d), the Mott gap like $1/R^{1/(1-g)}$, with predicted[47,50–52] values of $g$ pointing to a faster decay, $g < 1$. The EI order parameter is suppressed at high temperature (Fig. 5c) and strong magnetic field (Fig. 6c); the Mott gap is likely independent of both fields (the Aharonov–Bohm phase does not affect Hubbard-like Coulomb integrals). Importantly, the EI gap is very sensitive to the dielectric environment[69], whereas the Mott gap is not. This could explain the dramatic variation of narrow transport gaps of suspended tubes submerged in different liquid dielectrics[42].

We anticipate that armchair tubes exhibit an optical absorption spectrum in the THz range dominated by excitons, which

provides an independent test of the EI phase. Furthermore, we predict they behave as "chiral electronic ferroelectrics", displaying a permanent electric polarization **P** of purely electronic origin[7], whereas conventional ferroelectricity originates from ionic displacements. In fact, the volume average of **P** is zero but its circulation along the tube circumference is finite. Therefore, a suitable time-dependent field excites the ferroelectric resonance[7] associated with the oscillation of **P**. The special symmetry of armchair tubes[61] is expected to protect this collective (Goldstone) mode of oscillating electric dipoles from phase-locking mechanisms. The resulting soft mode—a displacement current along the tube circumference—is a manifestation of the long-debated[6–11,70,71] exciton superfluidity.

In conclusion, our calculations demonstrated that an isolated armchair carbon nanotube at charge neutrality is an EI, owing to the strong e–h binding in quasi-1D, and the almost unscreened long-range interactions. The emergence of this exotic state of matter, predicted fifty years ago, does not fit the common picture of carbon nanotubes as Luttinger liquids. Our first-principles calculations provide tests to discriminate between the EI and the Luttinger liquid at strong coupling, the Mott insulator state. We expect a wide family of narrow-gap carbon nanotubes to be EIs. Carbon nanotubes are thus invaluable systems for the experimental investigation of this phase of matter.

## Methods

**Many-body perturbation theory from first principles.** The ground-state calculations for the (3, 3) carbon nanotube were performed by using a DFT approach, as implemented in the Quantum ESPRESSO package[72]. The generalized gradient approximation (GGA) PW91 parametrization[73] was adopted together with plane wave basis set and norm-conserving pseudopotentials to model the electron–ion interaction. The kinetic energy cutoff for the wave functions was set to 70 Ry. The Brillouin zone was sampled by using a $200 \times 1 \times 1$ $k$-point grid. The supercell side perpendicular to the tube was set to 38 Bohr and checked to be large enough to avoid spurious interactions with its replica.

Many-body perturbation theory[44] calculations were performed using the Yambo code[74]. Many-body corrections to the Kohn–Sham eigenvalues were calculated within the $G0W0$ approximation to the self-energy operator, where the dynamic dielectric function was obtained within the plasmon-pole approximation. The spectrum of excited states was then computed by solving the BSE. The static screening in the direct term was calculated within the random-phase approximation with inclusion of local field effects; the Tamm–Dancoff approximation for the BSE Hamiltonian was employed after having verified that the correction introduced by coupling the resonant and antiresonant part was negligible. Converged excitation energies, $\varepsilon_u$, were obtained considering respectively three valence and four conduction bands in the BSE matrix. For the calculations of the GW band structure and the Bethe–Salpeter matrix the Brillouin zone was sampled with a $1793 \times 1 \times 1$ $k$-point grid. A kinetic energy cutoff of 55 Ry was used for the evaluation of the exchange part of the self energy and 4 Ry for the screening matrix size. Eighty unoccupied bands were used in the integration of the self-energy.

The effect of the magnetic field parallel to the axis on the electronic structure of the nanotube ground state (eigenvalues and eigenfunctions) was investigated following the method by Sangalli and Marini[67]. For each value of the field, the eigenvalues and eigenfunctions were considered to build the screening matrix and the corresponding excitonic Hamiltonian.

To obtain the equilibrium structure, we first considered possible corrugation effects. We computed the total energy for a set of structures obtained by varying the relative positions of A and B carbon atoms belonging to different sublattices, so that they were displaced one from the other along the radial direction by the corrugation length $\Delta$ and formed two cylinders, as in Fig. 1b of Lu et al.[60]. Then, we fitted the total energy per carbon atom with an elliptic paraboloid in the two-dimensional parameter space spanned by $\Delta$ and the carbon bond length. In agreement with Lu et al.[60], we find a corrugated structure with a bond length of 1.431 Å and a corrugation parameter $\Delta$ 0.018 Å. Eventually, starting from this structure, we performed a full geometry relaxation of the whole system allowing all carbon positions to change until the forces acting on all atoms became less than $5 \times 10^{-3}$ eV Å$^{-1}$. After relaxation, the final structure presents a negligible corrugation ($\Delta < 10^{-5}$ Å) and an average length of C–C bonds along the tube axis, 1.431 Å, slightly shorter than the C–C bonds around the tube circumference, 1.438 Å. The average radius and translation vector of the tube are respectively 2.101 and 2.462 Å, in perfect agreement with the literature[53]. The obtained equilibrium coordinates of C atoms in the unitary cell are shown in Supplementary Table 1.

**Quantum Monte Carlo method.** We have applied the quantum Monte Carlo method to carbon nanotubes by using standard pseudopotentials for the $1s$ core electrons of the carbon atom[75]. We minimize the total energy expectation value of the first-principles Hamiltonian, within the Born–Oppenheimer approximation, by means of a correlated wave function, $J|\text{SD}\rangle$. This is made of a Slater determinant, $|\text{SD}\rangle$, defined in a localized GTO VDZ basis[75] ($5s5p1d$) contracted into six hybrid orbitals per carbon atom[76], multiplied by a Jastrow term, $J$. The latter, $J = J_1 J_2$, is the product of two factors: a one-electron term, $J_1 = \prod_i \exp\left[u_{1\text{body}}(\mathbf{r}_i)\right]$, and a two-electron correlation factor, $J_2 = \prod_{i<j} \exp\left[u(\mathbf{r}_i, \mathbf{r}_j)\right]$. The two-body Jastrow factor $J_2$ depends explicitly on the $N_e$ electronic positions, $\{\mathbf{r}_i\}$, and, parametrically, on the $N_C$ carbon positions, $\mathbf{R}_I$, $I=1, \ldots N_C$. The pseudopotential functions, $u$ and $u_{1\text{body}}$, are written as:

$$u(\mathbf{r}, \mathbf{r}') = u_{ee}(|\mathbf{r} - \mathbf{r}'|) + \sum_{\mu>0,\nu>0} u_{\mu\nu} \chi_\mu(\mathbf{r}) \chi_\nu(\mathbf{r}'), \tag{4}$$

$$u_{1\text{body}}(\mathbf{r}) = \sum_{\mu>0} u_{\mu 0} \chi_\mu(\mathbf{r}), \tag{5}$$

where $u_{ee} = 2^{-1} r/(1 + b_{ee} r)$ is a simple function, depending on the single variational parameter $b_{ee}$, which allows to satisfy the electron–electron cusp condition, and $u_{\mu\nu}$ is a symmetric matrix of finite dimension. For non-null indices, $\mu, \nu > 0$, the matrix **u** describes the variational freedom of $J_2$ in a certain finite atomic basis, $\chi_\mu(\mathbf{r})$, which is localized around the atomic centers $\mathbf{R}_{I(\mu)}$ and is made of $3s2p$ GTO orbitals per atom. Note that the one-body Jastrow term $J_1$ is expanded over the same atomic basis and its variational freedom is determined by the first column of the matrix, $u_{\mu 0}$.

We use an orthorhombic unit cell $L_x \times L_y \times L_z$ containing twelve atoms with $L_x = L_y = 36$ Å and $L_z = 2.445$ Å. This cell is repeated along the $z$ direction for $n = 1, 2, 3, 4, 5, 6$ times, up to 72 carbon atoms in the supercell. Periodic images in the $x$ and $y$ directions are far enough that their mutual interaction can be safely neglected. Conversely, in the $z$ direction we apply twisted periodic boundary conditions and we integrate over that with a number $n_\theta$ of twists, $n_\theta = 80, 40, 30, 20, 20, 20$ for $n = 1, 2, 3, 4, 5, 6$, respectively, large enough to have converged results for each supercell.

The initial Slater determinant was taken by performing a standard LDA calculation. The molecular orbitals, namely their expansion coefficients in the GTO localized basis set, as well as the matrix **u** determining the Jastrow factor, were simultaneously optimized with well established methods developed in recent years[77,78], which allows us to consider up to 3000 independent variational parameters in a very stable and efficient way. Note that the two-body Jastrow term $J_2$ can be chosen to explicitly recover the EI mean-field wave function (1), as shown in Supplementary Discussion. After the stochastic optimization the correlation functions/order parameters can be computed in a simple way within VMC.

We also employ lattice-regularized diffusion Monte Carlo (LRDMC) within the fixed-node approximation, using a lattice mesh of $a_{\text{mesh}} = 0.2$ and $a_{\text{mesh}} = 0.4$ a.u., respectively, in order to check the convergence for $a_{\text{mesh}} \to 0$. The fixed-node approximation is necessary for fermions for obtaining statistically meaningful ground-state properties. In this case the correlation functions/order parameters, depending only on local (i.e., diagonal in the basis) operators, such as the ones presented in this work, are computed with the forward walking technique[79], which allows the computation of pure expectation values on the fixed-node ground state.

**Code availability.** Many-body perturbation theory calculations were performed by means of the codes Yambo (http://www.yambo-code.org/) and Quantum ESPRESSO (http://www.quantum-espresso.org), which are both open source software. Quantum Monte Carlo calculations were based on TurboRVB code (http://trac.sissa.it/svn/TurboRVB), which is available from S.S. upon reasonable request.

**Data availability.** The data that support the findings of this study are available from the corresponding author upon reasonable request.

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

## Acknowledgements

This work was supported in part by European Union H2020-EINFRA-2015-1 program under grant agreement No. 676598 project "MaX–Materials Design at the Exascale". S.S. acknowledges computational resouces provided through the HPCI System Research Project No. hp160126 on the K computer at RIKEN Advanced Institute for Computational Science. D.V., E.M. & M.R. acknowledge PRACE for awarding them access to the Marconi system based in Italy at CINECA (Grant No. Pra14_3622).

## Author contributions

M.R. and E.M. initiated this project, D.V., E.M. and M.R. designed a comprehensive strategy to tackle the instability problem by means of different methods, D.V. developed the many-body perturbation theory calculations and analysis, D.V. and D.S. optimized the Yambo code for the calculation in the presence of the magnetic field, S.S. and M.B. developed the quantum Monte Carlo calculations and analysis, M.R. developed the effective-mass theory and wrote the paper, all authors contributed to the analysis of data and critically discussed the paper.

## Additional information

**Competing interests:** The authors declare no competing financial interests.

