## [Peer Review File · Nature Communications]

Reviewers' comments:

Reviewer #1 (Remarks to the Author):

This paper addresses a fundamental question: Do electron correlation effects result in the formation of an energy gap for a specific class of carbon nanotubes (armchair), that by symmetry have crossing bands at the Fermi energy in the independent particle model. Historically, this question was originally framed much more generally. For example, in the classic paper by Jerome, Rice and Kohn (Ref. 4 here), can interactions lead to the formation of the so-called exciton insulator state in materials such as zero-gap semiconductors or semi-metals formed by band overlap. Another manifestation of the same fundamental physics, coming from a different regime, concerns whether in optically excited semiconductors, as the excitation density rises, does the resultant combination of excited electrons and holes behave as a two component plasma (metallic) or an excitonic condensate (excitonic insulator). This latter example is not mentioned here, but one can start from the old review by Rice (Solid State Physics, 32, 1, 1977) and work backwards and forwards to see the relevant literature. This problem has proved quite subtle and progress for specific classes of materials would be of broad interest. The carbon nanotube case is a promising example to consider since the combination of reduced dimensionality and reduced screening lead to very strong effects from Coulomb interactions, e.g., the by now well established large magnitude of exciton binding energy for those classes of carbon nanotubes that are semiconducting. Unfortunately, I conclude that this paper does not present well-supported conclusions for the main part of the work, namely that based on quantum Monte Carlo calculations. In fairness, the necessary additional calculations may not be trivial, but a well-supported conclusion requires substantial additional work. I recommend rejection.

This paper starts with a well-executed set of calculations directed to an initial look at the problem using perturbation theory. The GW method together with BSE calculations are used to show stabilization of bound electron hole pairs with negative energy. To reiterate, these are fundamentally perturbation theory based calculations, so this is a physically interesting signal that there could be an instability towards the excitonic insulator phase, but it does not establish that this phase is indeed lower in energy.

For that purpose, the authors nominally turn to the use of quantum Monte Carlo calculations, initially using the variational method and then extending to a diffusion Monte Carlo method. Here, I find that the authors use a fundamentally flawed approach. They build their variational wavefunction with a Slater determinant that encodes the zero-band gap state and then apply standard Jastrow factors to capture correlation. While they see some residual correlations indicative of those expected in the excitonic insulator, they largely vanish in the thermodynamic limit, which I argue is exactly what one should expect. The state represented does not encode the excitonic insulator phase from the beginning and there is no reason to expect generic Jastrow-based correlations to bring it forth spontaneously. Instead, I should have expected that this calculation would have been the first one of a pair, establishing the energy of the 'normal' phase. What is completely missing is the other calculation in which the Slater determinant encodes the excitonic insulator phase (through the mean field theory as done by Keldysh and Kopaev and executed here separately) and to which correlations are added through an appropriate Jastrow factor. This would then give the energy estimate for the excitonic insulator phase. Which one is lower in energy? In short, the authors have only done half of the job.

To be absolutely clear, this is not a minor point. To establish that armchair carbon nanotubes are physically described by an excitonic insulator phase, the gold-standard is to convincingly argue that this phase has the lowest energy.

As an example of the procedure I am describing, the authors can refer to De Palo, Rapisarda and Senatore [PRL 88, 206401, 2002] who specifically use diffusion Monte Carlo to study the relative stability of the excitonic condensate relative to the electron-hole plasma and Wigner phases, for a

specific, quantum-well-based system, right down to the use of the Keldysh and Kopaev mean field theory to develop the input in the excitonic condensate phase.

In fairness to the complexity of doing full calculations considering all the carbon valence electrons (not just a reduced model Hamiltonian), the proper way to incorporate the mean-field theory results may be non-trivial to execute, although the relatively clear separation of energy scales between the bands that cross at the Fermi energy and other degrees of freedom should make it plausible.

I observe that in the expansive reference list on this paper, the exciton condensate side of the field is completely neglected. More attention to it would both have been appropriate for the reader and likely could have been informative regarding the approach as well.

I also note that, as I was sorting through recent, related literature, that there are also studies based on field theory techniques that in principle address similar points for graphene. This is intellectually connected and likely should have been cited as well.

Reviewer #2 (Remarks to the Author):

This is a very interesting paper, which introduces the possibility of the realization of a new state of matter (Excitonic Insulator - EIs) in carbon nanotubes. I believe this is an important contribution, which merits consideration for publication in Nature Communications.

The authors use two completely different methods to provide evidence of the possibility that "metallic" carbon nanotubes may actually be EIs in their ground state: many body perturbation theory for the calculation of the excitonic binding energy (demonstrating that it is negative), and Quantum Monte Carlo calculations of the ground state wave functions, showing the charge modulation expected for the EI.

I have two issues which I would like the authors to consider before recommending publication.

The first one is the definition of the order parameter for the Peierls CDW state in pg. 9. In my opinion, the order parameter as defined here is appropriate for a CDW with $q=\pi/a$; that is, a periodicity of two lattice vectors. This would be the case if the band structure of the nanotube was such that the Fermi level lay at $k=\pi/2a$ (ie, in the middle of the Gamma-M segment). However, this is not the case in the armchair CNTs considered (see fig. 2a). There, one would expect a Peierls CDW to occur for $q=2k_F$. If that was the case, then the order parameter as defined in pg. 9 would approach to zero, even if the system did have a $q=\pi/2a$ CDW. Can the authors comment on this?

The second one concerns the experimental evidence that has traditionally supported the image of CNTs as Luttinger liquids. To a large extent, this picture was supported mainly by transport experiments (for instance, regarding the dependence of the conductance vs temperature). The authors might try to provide some comments (even if qualitative) as how would the EI model explain (or not) this abundant literature.

Reply to Reviewer #1

This paper addresses a fundamental question: Do electron correlation effects result in the formation of an energy gap for a specific class of carbon nanotubes (armchair), that by symmetry have crossing bands at the Fermi energy in the independent particle model. Historically, this question was originally framed much more generally. For example, in the classic paper by Jerome, Rice and Kohn (Ref. 4 here), can interactions lead to the formation of the so-called exciton insulator state in materials such as zero-gap semiconductors or semi-metals formed by band overlap. Another manifestation of the same fundamental physics, coming from a different regime, concerns whether in optically excited semiconductors, as the excitation density rises, does the resultant combination of excited electrons and holes behave as a two component plasma (metallic) or an excitonic condensate (excitonic insulator). This latter example is not mentioned here, but one can start from the old review by Rice (Solid State Physics, 32, 1, 1977) and work backwards and forwards to see the relevant literature.

We thank Reviewer #1 for this observation that broadens the interest of our work. We have now expanded the reference list by including the review mentioned by Reviewer #1 as well as a more recent one by Keldysh.

This problem has proved quite subtle and progress for specific classes of materials would be of broad interest. The carbon nanotube case is a promising example to consider since the combination of reduced dimensionality and reduced screening lead to very strong effects from Coulomb interactions, e.g., the by now well established large magnitude of exciton binding energy for those classes of carbon nanotubes that are semiconducting.

We thank Reviewer #1 for acknowledging the broad interest of our paper.

Unfortunately, I conclude that this paper does not present well-supported conclusions for the main part of the work, namely that based on quantum Monte Carlo calculations. In fairness, the necessary additional calculations may not be trivial, but a well-supported conclusion requires substantial additional work. I recommend rejection.

We believe that the obscurity of the technical notes in the original version of the manuscript has led Reviewer #1 to misunderstand our computational approach. We put a significant effort into this revised version to clarify that the QMC calculation already contains all ingredients required by Reviewer #1. We are confident that our expanded discussion, which we detail below, will convince Reviewer #1 that our conclusions are well supported as well as no other calculations are needed.

This paper starts with a well-executed set of calculations directed to an initial look at the problem using perturbation theory. The GW method together with BSE calculations are used to show stabilization of bound electron hole pairs with negative energy. To reiterate, these are fundamentally perturbation theory based calculations, so this is a physically interesting signal that there could be an instability towards the excitonic insulator phase, but it does not establish that this phase is indeed lower in energy.

We would say instead that the solution of the Bethe-Salpeter equation is inherently non-perturbative, though limited to the one-exciton problem.

For that purpose, the authors nominally turn to the use of quantum Monte Carlo calculations, initially using the variational method and then extending to a diffusion Monte Carlo method. Here, I find that the authors use a fundamentally flawed approach. They build their variational wavefunction with a Slater determinant that encodes the zero-band gap state and then apply standard Jastrow factors to capture correlation.

The variational Monte Carlo (VMC) approach we used is non standard and differs from the algorithm summarized by Reviewer #1. We understand the Methods section we wrote was not helpful and we apologize for that. The key point is that the Jastrow factor is the product of two terms, the one-body factor and the two-body factor. **Both** are **optimized independently**, the space of variational degrees of freedom being huge (around 3000 independent parameters). This is of the utmost importance with regard to the critique of Reviewer #1, since the separate optimization of the one-body factor is equivalent to an almost arbitrary unitary transformation of the single-particle basis set. Therefore, apart from the standard two-body Jastrow factor, our VMC algorithm is able to turn the zero-gap Slater determinant into the mean-field (Keldysh-Kopaev) EI wave function. In fact the latter, written in terms of e-h pair creation operators acting on the zero-gap Slater determinant, $|0\rangle$ (see e.g. Ref. 11),

$$|\Psi_{\text{EI}}\rangle = \prod_k (u_k + v_k \hat{c}_k^+ \hat{v}_k) |0\rangle,$$

may equivalently be written in standard Slater determinant form,

$$|\Psi_{\text{EI}}\rangle = \prod_k (u_k \hat{v}_k^+ + v_k \hat{c}_k^+) |\text{vac}\rangle,$$

where $|\text{vac}\rangle$ is the true vacuum and the single-particle orbitals $(u_k \hat{v}_k^+ + v_k \hat{c}_k^+)$ are obtained through a proper rotation of the DFT single-particle basis set (see e.g. Eq. 2.9 of Ref. 4; for the sake of clarity in this reply we have neglected spin and valley degrees of freedom), which is fully accessible by our QMC simulation. The substance of our approach is now clarified in the Methods section.

Moreover, we have added a detailed Supplementary Discussion, ‘The EI mean-field wave function as specialization of the QMC variational wave function’ (pages 35-39), where we rigorously demonstrate that the Jastrow factor is able to recover the Keldysh-Kopaev form of the ansatz wave function.

In summary, our VMC approach may be thought of as the following sequence: zero-gap Slater determinant \rightarrow Keldysh-Kopaev wave function \rightarrow Jastrow-correlated wave function, the optimizations related to one- and two-body terms running independently. We have now emphasized in the main text that our VMC method has the ability to fully include the excitonic correlations built in the Keldysh-Kopaev ansatz. There is no reason to expect that our approach would provide different results if it started from the Keldysh-Kopaev wave function instead of the zero-gap state, as done in the paper mentioned by Reviewer #1, De Palo, Rapisarda & Senatore 2002 (new reference 25).

While they see some residual correlations indicative of those expected in the excitonic insulator, they largely vanish in the thermodynamic limit, which I argue is exactly what one should expect.

The mean-field prediction (à la Keldysh-Kopaev) for the charge displacement between sublattices ($\rho_{AB} = 0.0068$) compares well with our QMC findings ($\rho_{AB} = 0.0067$ and 0.0165 from LRDMC and VQMC, respectively). Therefore, QMC in the thermodynamic limit recovers all mean-field excitonic correlations, i.e., **no larger** excitonic correlations may sensibly be expected. Though, the broken inversion symmetry of the EI phase profoundly affects observable properties such as the quasiparticle gap and the critical temperature, which are at least one order of magnitude larger than the corresponding properties of superconductors well described by BCS theory, such as niobium.

The state represented does not encode the excitonic insulator phase from the beginning and there is no reason to expect generic Jastrow-based correlations to bring it forth spontaneously.

Above we have shown that our VMC approach has the full variational power to include those excitonic correlations built in the Keldysh-Kopaev wave function. In addition, we note that the excitonic insulator ground state resembles a charge density wave (CDW), with the important exception that the breaking of sublattice symmetry has a purely electronic origin. It is shown in the literature that our QMC method well fits to the CDW problem [M. Capello *et al.*, PRL **94**, 026406 (2005); PRB **72**, 085121 (2005); Tayo & Sorella, PRB **78**, 115117 (2008)], including evidence that the Jastrow factor spontaneously brings forth CDW-like order [Kaneko *et al.*, PRB **93**, 125127 (2016)].

Instead, I should have expected that this calculation would have been the first one of a pair, establishing the energy of the 'normal' phase. What is completely missing is the other calculation in which the Slater determinant encodes the excitonic insulator phase (through the mean field theory as done by Keldysh and Kopaev and executed here separately) and to which correlations are added through an appropriate Jastrow factor. This would then give the energy estimate for the excitonic insulator phase. Which one is lower in energy? In short, the authors have only done half of the job.

To be absolutely clear, this is not a minor point. To establish that armchair carbon nanotubes are physically described by an excitonic insulator phase, the gold-standard is to convincingly argue that this phase has the lowest energy.

As an example of the procedure I am describing, the authors can refer to De Palo, Rapisarda and Senatore [PRL 88, 206401, 2002] who specifically use diffusion Monte Carlo to study the relative stability of the excitonic condensate relative to the electron-hole plasma and Wigner phases, for a specific, quantum-well-based system, right down to the use of the Keldysh and Kopaev mean field theory to develop the input in the excitonic condensate phase.

In fairness to the complexity of doing full calculations considering all the carbon valence electrons (not just a reduced model Hamiltonian), the proper way to incorporate the mean-field theory results may be non-trivial to execute, although the relatively clear separation of energy scales between the bands that cross at the Fermi energy and other degrees of freedom should make it plausible.

As discussed above, the method by De Palo and coworkers (new reference 25) cannot add anything new to our present computational capability. Of course, we agree with Reviewer #1 that the gold-standard would be to compare the EI ground state energy with that of the normal phase. To this aim, we have checked that the mean-field energy gain of the excitonic insulator with respect to the zero-gap solution (below 10^{-6} Hartree per atom) is less than the lowest QMC noise threshold possible (10^{-5} Hartree per atom). Therefore, further calculations are pointless. Indeed, in a Monte Carlo calculation a small order parameter can be detected by an appropriate finite-size scaling, where each individual finite-size simulation allows fluctuations between small positive and negative values of the broken-symmetry order parameter. The signature of the EI phase is therefore unambiguous if the square order parameter is finite in the thermodynamic limit, as we have shown in our work.

I observe that in the expansive reference list on this paper, the exciton condensate side of the field is completely neglected. More attention to it would both have been appropriate for the reader and likely could have been informative regarding the approach as well.

We have expanded our reference list to take Reviewer #1's comments into account, adding Ref. 25 as well as two recent reviews on exciton condensates (Refs. 13 and 26). We would like to emphasize that references from 1 to 38 concern exciton condensates. We have tried our best to fairly acknowledge the vast literature.

I also note that, as I was sorting through recent, related literature, that there are also studies based on field theory techniques that in principle address similar points for graphene. This is intellectually connected and likely should have been cited as well.

In our original version we had cited a seminal field-theoretical work on graphene dating back to 2001 (Ref. 27), as well as Refs. 28-30 on the same topic. We have now added a recent review on related theoretical work in graphene (Ref. 31).

Reply to Reviewer #2

This is a very interesting paper, which introduces the possibility of the realization of a new state of matter (Excitonic Insulator - EIs) in carbon nanotubes. I believe this is an important contribution, which merits consideration for publication in Nature Communications.

The authors use two completely different methods to provide evidence of the possibility that "metallic" carbon nanotubes may actually be EIs in their ground state: many body perturbation theory for the calculation of the excitonic binding energy (demonstrating that it is negative), and Quantum Monte Carlo calculations of the ground state wave functions, showing the charge modulation expected for the EI.

We thank Reviewer #2 very much for the warm appreciation of our work.

I have two issues which I would like the authors to consider before recommending publication.

The first one is the definition of the order parameter for the Peierls CDW state in pg. 9. In my opinion, the order parameter as defined here is appropriate for a CDW with $q=\pi/a$; that is, a periodicity of two lattice vectors. This would be the case if the band structure of the nanotube was such that the Fermi level lay at $k=\pi/2a$ (ie, in the middle of the Gamma-M segment). However, this is not the case in the armchair CNTs considered (see fig. 2a). There, one would expect a Peierls CDW to occur for $q=2k_F$. If that was the case, then the order parameter as defined in pg. 9 would approach to zero, even if the system did have a $q=\pi/2a$ CDW. Can the authors comment on this?

Reviewer #2 is right. The order parameter ρ_{Transl} is finite in the thermodynamic limit only if the CDW wave vector is $q = \pi / a$. However, this same order parameter, evaluated over a generic supercell for a Peierls CDW with $q = 2 k_F$, is finite and provides significant information even if it vanishes as $1/N_{\text{atom}} \rightarrow 0$. This conclusion we reach by observing that, for a simple model of Peierls CDW, the finite-size scaling of $(\rho_{\text{Transl}})^2$ is non-monotonous and it largely departs from the linear scaling observed in the QMC simulation. We have now added this argument in the revised Supplementary Discussion ‘Detection of Peierls charge density wave through the order parameter ρ_{Transl} ’ and Supplementary Figure 12. In addition, we have clarified in the main text (and caption of Fig. 4) that the QMC analysis only concerns the dimerized CDW with $q = \pi / a$, referring the reader interested in the Peierls CDW to the Supplementary Discussion as well as the pertinent theoretical literature.

The second one concerns the experimental evidence that has traditionally supported the image of CNTs as Luttinger liquids. To a large extent, this picture was supported mainly by transport experiments (for instance, regarding the dependence of the conductance vs temperature). The authors might try to provide some comments (even if qualitative) as how would the EI model explain (or not) this abundant literature.

The experimental evidence of Luttinger liquid behaviour concerns doped carbon nanotubes. Such evidence hardly confutes the EI picture we put forward since, by doping the tube with electrons or holes, one expects the enhanced screening to effectively suppress the EI phase, turning it into the Luttinger liquid. Importantly, a few free carries injected into the tube are sufficient to destroy the long range of Coulomb interaction, as shown e.g. in the experiment on suspended ultraclean tubes by the Bockrath group reported in *Nature Phys.* **4**, 314 (2008). As demonstrated in our manuscript, long-range interaction is pivotal to the excitonic instability.

To clarify this issue, the revised text now contains the following expanded paragraph:

‘In experiments, many-body gaps are observed in undoped, ultraclean suspended tubes⁷², whereas Luttinger liquid signatures emerge in doped tubes^{40,41}. Though it is difficult to compare with the measured many-body gaps⁷², as the chiralities of the tubes are unknown and the radii estimated indirectly, the measured range of 10–100 meV is at least one order of magnitude larger than our predictions. By doping the tube, we expect that the enhanced screening suppresses the EI order, quickly turning the system into a Luttinger liquid. We are confident that advances in electron spectroscopies will allow to test our theory.’

REVIEWERS' COMMENTS:

Reviewer #1 (Remarks to the Author):

I have reviewed the reply of the authors and the changes to the manuscript and SI. My primary concern about the original manuscript centered on the trial wave function for the Monte Carlo calculations. We can agree that the original description was not clear! In the revised manuscript, my concerns have largely been addressed. I have some specific suggestions below which I think are important, but should be easy to satisfy. With those matters addressed, I can strongly recommend this paper for publication. The scope of the study, addressing a fundamentally interesting problem, is to be commended.

The authors address the point regarding the energy difference between the EI phase and the semimetal phase in their reply, but not in the manuscript. I think this is an important point and it should be covered in the paper itself with a brief paragraph. The reader should know that the authors have examined this point. The fact that the energy difference is below statistical resolution is also useful information and can be physically related to the temperature scale that emerges in the mean-field model calculations.

In the methods section, there should be a sentence that explicitly states that the form of $u(r,r')$ chosen can be shown to encompass the usual form of the trial function for the EI phase and then refer to the extensive discussion in the SI. The analysis in the SI is good, but the reader needs a guide to link these things together.

The comment directed to the two-leg Hubbard ladder on page 11 ("no order would set in the thermodynamic limit") does not make sense -- I can not quite guess what the authors mean. This needs to get fixed.

Reviewer #2 (Remarks to the Author):

In my view, the authors have responded satisfactorily to the comments of my report, and (as far as I understand) to the ones of the other referee.

In particular, my question about the relevance of the order parameter for the $\pi/2a$ CDW case for the present system (in which k_f is not $\pi/2a$), is convincing. Although this quantity does not provide a direct proof of the existence of the CDW state, it is an indication which gives support to the claims of the paper.

The comments about the connection with the Luttinger state (relevant for doped systems) are also satisfactory.

Concerning the main remark of the other referee (the form of the VMC wavefunction), the former text was not sufficiently clear, but I believe that the current one satisfies what the referee was requesting.

In brief, I think the paper is sound, and the authors have replied satisfactorily to our concerns. The current version of the paper can now be accepted for publication.

Reply to Reviewer #1

I have reviewed the reply of the authors and the changes to the manuscript and SI. My primary concern about the original manuscript centered on the trial wave function for the Monte Carlo calculations. We can agree that the original description was not clear! In the revised manuscript, my concerns have largely been addressed. I have some specific suggestions below which I think are important, but should be easy to satisfy. With those matters addressed, I can strongly recommend this paper for publication. The scope of the study, addressing a fundamentally interesting problem, is to be commended.

We thank Reviewer #1 for appreciating our work, as well as for the suggestions mentioned below.

The authors address the point regarding the energy difference between the EI phase and the semimetal phase in their reply, but not in the manuscript. I think this is an important point and it should be covered in the paper itself with a brief paragraph. The reader should know that the authors have examined this point. The fact that the energy difference is below statistical resolution is also useful information and can be physically related to the temperature scale that emerges in the mean-field model calculations.

We have now added the following sentence at the end of the paragraph discussing the charge displacement between sublattices (last paragraph before section ‘Effect of magnetic field’): ‘Note that assessing the energy difference between EI and zero-gap ground states is beyond the current capability of quantum Monte Carlo: the mean-field estimate of the difference is below 10^{-6} Hartree per atom, which is less than the noise threshold of the method (10^{-5} Hartree per atom)’.

In the methods section, there should be a sentence that explicitly states that the form of $u(r,r')$ chosen can be shown to encompass the usual form of the trial function for the EI phase and then refer to the extensive discussion in the SI. The analysis in the SI is good, but the reader needs a guide to link these things together.

We have now added the following sentence in the Methods section on quantum Monte Carlo: ‘Note that the two-body Jastrow term J_2 can be chosen to explicitly recover the EI mean-field wave function (1), as shown in Supplementary Discussion.’

The comment directed to the two-leg Hubbard ladder on page 11 ("no order would set in the thermodynamic limit") does not make sense -- I can not quite guess what the authors mean. This needs to get fixed.

We have removed the comment.

Reply to Reviewer #2

In my view, the authors have responded satisfactorily to the comments of my report, and (as far as I understand) to the ones of the other referee.

In particular, my question about the relevance of the order parameter for the $\pi/2a$ CDW case for the present system (in which k_f is not $\pi/2a$), is convincing. Although this quantity does not provide a direct proof of the existence of the CDW state, it is an indication which gives support to the claims of the paper.

The comments about the connection with the Luttinger state (relevant for doped systems) are also satisfactory.

Concerning the main remark of the other referee (the form of the VMC wavefunction), the former text was not sufficiently clear, but I believe that the current one satisfies what the referee was requesting.

In brief, I think the paper is sound, and the authors have replied satisfactorily to our concerns. The current version of the paper can now be accepted for publication.

We thank Reviewer #2 for recommending our work for publication.